# Measuring Canopy Structure and Condition Using Multi-Spectral UAS Imagery in a Horticultural Environment

**Yu-Hsuan Tu** [1,2,*] , **Kasper Johansen** [1,3] , **Stuart Phinn** [1,2] and **Andrew Robson** [2,4]

[1]   Remote Sensing Research Centre, School of Earth and Environmental Sciences,
     The University of Queensland, St Lucia 4072, QLD, Australia; kasper.johansen@kaust.edu.sa (K.J.);
     s.phinn@uq.edu.au (S.P.)
[2]   Joint Remote Sensing Research Program, The University of Queensland, St Lucia 4072, QLD, Australia
[3]   Hydrology, Agriculture and Land Observation Group, Water Desalination and Reuse Center,
     King Abdullah University of Science and Technology, Thuwal 23955-6900, Saudi Arabia
[4]   Precision Agriculture Research Group, School of Science and Technology, University of New England,
     Armidale 2351, NSW, Australia; andrew.robson@une.edu.au
*   Correspondence: y.tu@uq.edu.au; Tel.: +61-7-3346-7023

**Abstract:** Tree condition, pruning and orchard management practices within intensive horticultural tree crop systems can be determined via measurements of tree structure. Multi-spectral imagery acquired from an unmanned aerial system (UAS) has been demonstrated as an accurate and efficient platform for measuring various tree structural attributes, but research in complex horticultural environments has been limited. This research established a methodology for accurately estimating tree crown height, extent, plant projective cover (PPC) and condition of avocado tree crops, from a UAS platform. Individual tree crowns were delineated using object-based image analysis. In comparison to field measured canopy heights, an image-derived canopy height model provided a coefficient of determination ($R^2$) of 0.65 and relative root mean squared error of 6%. Tree crown length perpendicular to the hedgerow was accurately mapped. PPC was measured using spectral and textural image information and produced an $R^2$ value of 0.62 against field data. A random forest classifier was applied to assign tree condition into four categories in accordance with industry standards, producing out-of-bag accuracies >96%. Our results demonstrate the potential of UAS-based mapping for the provision of information to support the horticulture industry and facilitate orchard-based assessment and management.

**Keywords:** unmanned aerial system; horticulture; avocado; canopy structure; tree condition

## 1. Introduction

Genetic variation, phenological growth stage and abiotic and biotic constraints can all influence tree structural parameters such as height, canopy extent and foliage density [1–12]. For instance, vigorous trees usually result in tall and dense canopies that influence the productivity [9–11]. As such, measurements of these parameters can provide growers with a strong indication of plant health or vigour, photosynthetic capacity and yield potential [13,14]. In most Australian horticultural industries, such assessment is usually conducted by on-ground visual evaluation, which is time-consuming, labour-intensive, subjective and often inconsistent [7,15–17]. Therefore, there is a demand for more efficient, accurate and quantitative alternatives for such assessments.

Remote sensing has been used to estimate tree canopy structural attributes, such as tree height, canopy extent, and plant projective cover (PPC), for decades [18–22]. Conventionally,

these types of data were acquired by either satellites or aircrafts, which are time and weather constrained. In the case of satellite data, it is difficult to acquire data with on-demand spatial, spectral, and temporal specifications that is tailored for specific sites, products, and delivery times for horticultural applications [23]. More recently, UAS have been considered as a useful platform to acquire suitable remotely sensed data for measuring horticultural tree crop structure, as it provides both higher spatial and temporal resolution and the flexibility of operations [6–8,18]. Airborne image data, initially stereo-photography, were analyzed using soft-copy photogrammetry to provide information for extracting ground and canopy heights [2–7,24], while more recently this has been done by collection and processing of LiDAR point clouds [1,25–28]. Similar products are now generated from UAS multi-band image data using multiple forms of softcopy photogrammetry, such as structure from motion (SfM). As SfM algorithms have become part of the standard procedure of UAS image processing workflows, UAS imagery has the potential to become a more accepted method for measuring tree structure, providing a similar degree of accuracy and better spectral fusion compared to LiDAR systems [6,29].

Canopy structural dimensions can be expressed by the tree height and the tree crown extent [4], while the canopy density can be expressed by the PPC [21]. PPC is defined as the vertically projected fraction of leaves and branches in relation to sky, which is often referred to as canopy cover or crown cover [20–22]. It has been proved to be a good indicator of biomass [20,21]. Previous studies have estimated PPC at the site scale [22,30]. However, for tree crop managers, the plant-scale condition of individual trees is more relevant when applying agricultural inputs at the tree level [16]. Plant-scale PPC estimation using near-infrared (NIR) brightness from multi-spectral UAS imagery has proved to be an accurate approach for lychee trees [7]. As the canopy structure of lychee trees is considered to be 'rhythmic' [31], little is known about whether such plant-scale PPC estimation techniques are feasible for other tree crops such as avocado trees, as the architecture of avocado trees is significantly different to lychee trees [32]. Although Salgadoe, et al. [33] suggested that photo-derived PPC is also a good indicator of avocado tree condition, the difference of canopy density is not significant to differentiate trees between moderate and poor condition [33]. Hence, combining the canopy structural dimension for tree vigour estimation may be beneficial [13]. The estimated tree height and canopy extent derived from UAS imagery are now considered highly accurate in comparison to ground measurements [4–8], yet limited research has integrated the dimension and PPC to express tree condition.

Tree condition evaluation methods are generally conducted by classifying trees into condition categories based on visual assessment of tree canopy structure. This assessment is based on tree canopy structure, and hence should be quantifiable from the aforementioned structural attributes that derived from UAS imagery. Johansen, Duan, Tu, Searle, Wu, Phinn and Robson [15] used near-infrared (NIR) brightness as the input feature of a random forest classifier to predict the condition of macadamia trees. Different growing stages and levels of stress for avocado trees produce changes in overall canopy form [33], and the amount of leaf, stem and trunk biomass, these resulted in different NIR reflectance levels [34–36] which were detected by the classifier in the study. The results indicated a moderate accuracy, hypothesised to be attributed to the very high spatial resolution of the UAS imagery, combined with the complex NIR scattering properties of the tree crop leaf and canopy structures. Based on this foundation, we explored methods that use all the available image-derived structural attributes and linked these with the field-derived structural measurements for condition assessment.

The objectives of this work were to:

1. use multi-spectral UAS imagery to map and assess the accuracy of tree height, canopy width and length, and PPC estimates for avocado trees, and
2. use the highly correlated image-derived structural attribute maps to produce a tree condition ranking matched to on-round observer techniques.

This study explores a novel and innovative approach to assess horticultural tree crop condition directly related to canopy structural attributes estimated from UAS image data and used to rank

avocado tree condition. Such a condition ranking strategy could play a role to bridge the gap between the remote sensing expertise and farmers' knowledge of their tree crops, delivering information able to be used immediately as part of on farm management practices.

## 2. Materials and Methods

### 2.1. Study Sites

The study site is one of the main Hass avocado producing orchards in Australia, which is located to the south of Bundaberg, Queensland, Australia. The region experiences a subtropical climate with long hot summers and mild winters. The post-harvest pruning usually occurs in late Austral winter (August) or early spring (September), where major limbs are removed to maintain orchard access between rows and enhance light interception. Flowering occurs between early September and mid-October, followed by two fruit drop events usually between late October and January for readjusting fruit load to fit the tree ecological resources [37]. Harvesting of avocado normally occurs in May. The UAS based focus areas encompassed a 1 ha patch of the avocado orchard, within which structural parameters of 45 avocado trees were evenly selected and measured (Figure 1). The avocado trees were planted in 2005 with a 5 m spacing. The average elevation for the avocado orchard was around 60 m above sea level, and it had relatively flat terrain with an average downward slope of 4 degrees towards east.

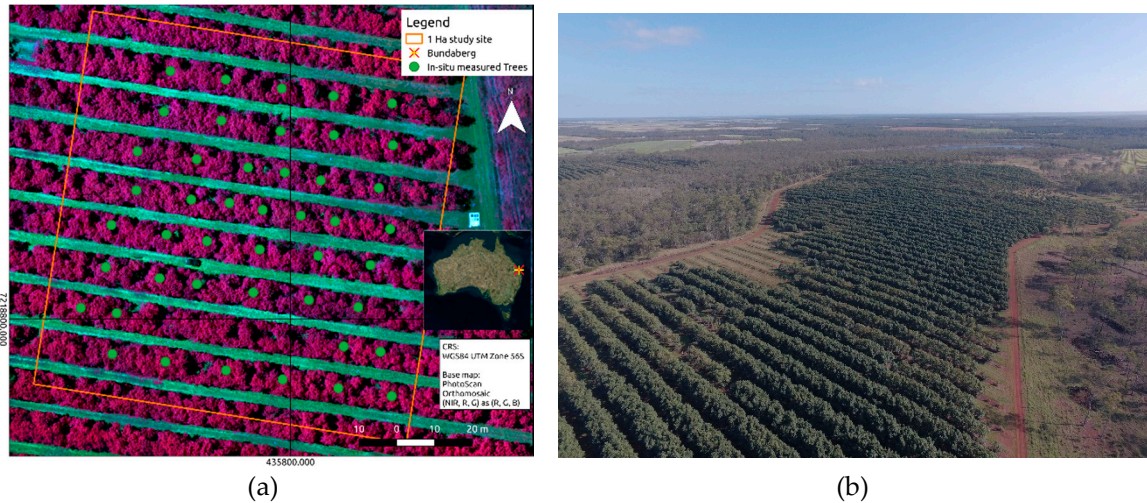

(a)　　　　　　　　　　　　　　　　　　　　　　　　　　(b)

**Figure 1.** The (**a**) outline of the study area and (**b**) aerial view of the avocado orchard in Bundaberg, Australia. The green dots in (**a**) represent the selected in-situ measured trees. The Australia basemap in (**a**) uses imagery from World-View 2 provided by DigitalGlobe.

### 2.2. Field Data

Field measurements were collected between 2 and 6 February 2017, where the canopy becomes dense at the end of summer leave flush [38]. Tree height was measured using a TruPulse 360B laser rangefinder (Laser Technology Inc, Centennial, USA) from the ground to the tree apex at a distance greater than 10 m from the trunk to reduce the effect of tree inclination [39]. Canopy widths were measured, in the direction along the hedgerow, as the horizontal distance between the two staffs placed at the outermost edge of each side of the tree crown.

PPC for individual trees was estimated via a digital cover photography (DCP) method [21] using a Nikon Coolpix AW120 digital camera (Nikon Corporation, Tokyo, Japan). Eight representative cover images were taken at various locations at approximately half way between the trunk and the outer edge of the tree crown of each tree during dusk. These images were imported into the Can-Eye

software (French National Institute of Agronomical research, Paris, France) to estimate gap fraction to calculate PPC.

A total of 66 trees, selected based on their variability in condition, were visually assessed by an avocado tree expert and classified into four condition categories: (1) Excellent, (2) Good, (3) Moderate, and (4) Fair (Figure 2), based on the modified Ciba-Geigy scale method for the Australian avocado industry [33]. Twenty-two of the 66 assessed trees were part of the selected 45 trees, which had their structural properties measured for validation purposes. The number of trees for each condition category can be found in Table 1.

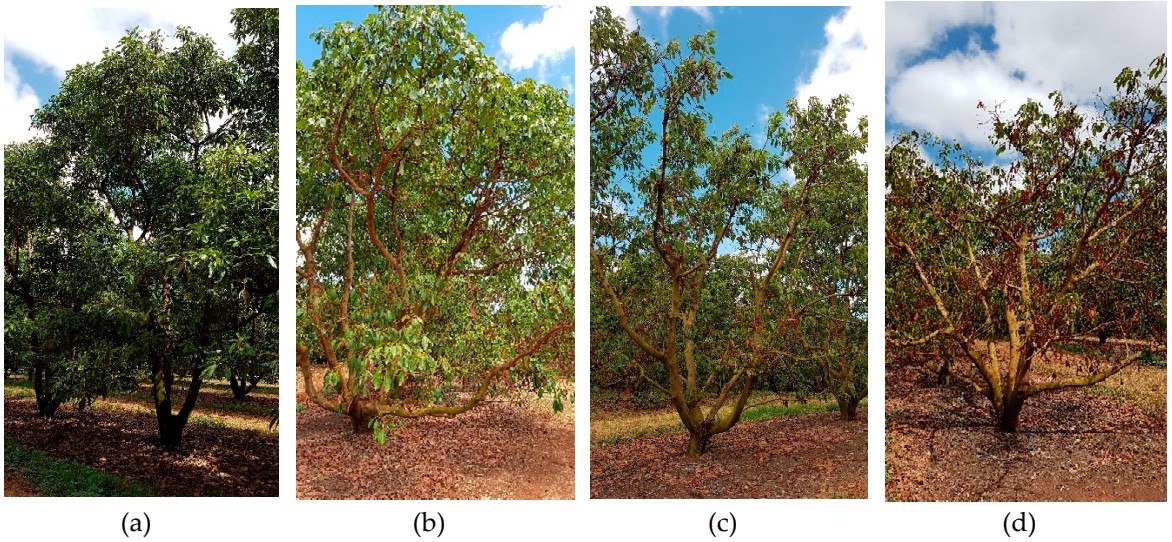

|        (a)        |        (b)        |        (c)        |        (d)        |

**Figure 2.** The condition ranking for avocado trees based on farmers' knowledge. The standard ranking method classifies the trees into four categories: (**a**) Excellent, where there was no sign of defoliation; (**b**) Good, with few wilting leaves occurring; (**c**) Moderate, where approximate one third to half of the green leaf tissues were wilted; and (**d**) Fair, where more than half of the leaves were dead.

**Table 1.** Number of avocado trees classified in situ into respective condition rankings.

| Ranking | 1- Excellent | 2- Good | 3- Moderate | 4- Fair |
|---------|--------------|---------|-------------|---------|
| **Count** | 8 | 21 | 32 | 5 |

*2.3. UAS Data Acquisition*

The multi-spectral UAS imagery was captured with a Parrot Sequoia®multi-spectral camera (Parrot Drone SAS, Paris, France) mounted on a 3DR Solo (3D Robotics, Berkeley, USA) quadcopter under clear sky conditions. The camera acquires imagery in the green (550 nm, 40 nm bandwidth), red (660 nm, 40 nm bandwidth), red edge (735 nm, 10 nm bandwidth), and NIR (790 nm, 40 nm bandwidth) part of the spectrum with its 4.8 × 3.6 mm (1280 × 960 pixels) CMOS sensor. The image acquisition parameters, i.e., flight height, sidelap, and flight speed, etc., were designed based on the suggestions from previous studies [7,40,41]. The flight was conducted in an along-tree-row pattern at 75 m AGL on 2 Feb 2017 between 2:11-2:17 PM (60° solar elevation). The sidelap of the images was 80%, the image capture interval was set to 1 second (92% forward overlap), and the flight speed was 5 m/s.

Ten AeroPoints®(Propeller Aerobotics Pty Ltd, Surry Hills, Australia) and eight gradient panels in greyscale were deployed for geometric and radiometric correction purposes, respectively. The locations of the AeroPoints were recorded for five hours and subsequently post-processed using the Propeller®network correction based on the nearest base station, located within 26 km of the study site. The calibration panels were designed based on the suggestion of Wang and Myint [42]. The reflectance

values of the calibration panels were measured with an ASD FieldSpec®3 spectrometer (Malvern Panalytical Ltd, Malvern, UK) and ranged from 4% to 92% in all spectral bands, corresponding with the four bands of the Sequoia®camera (Figure 3).

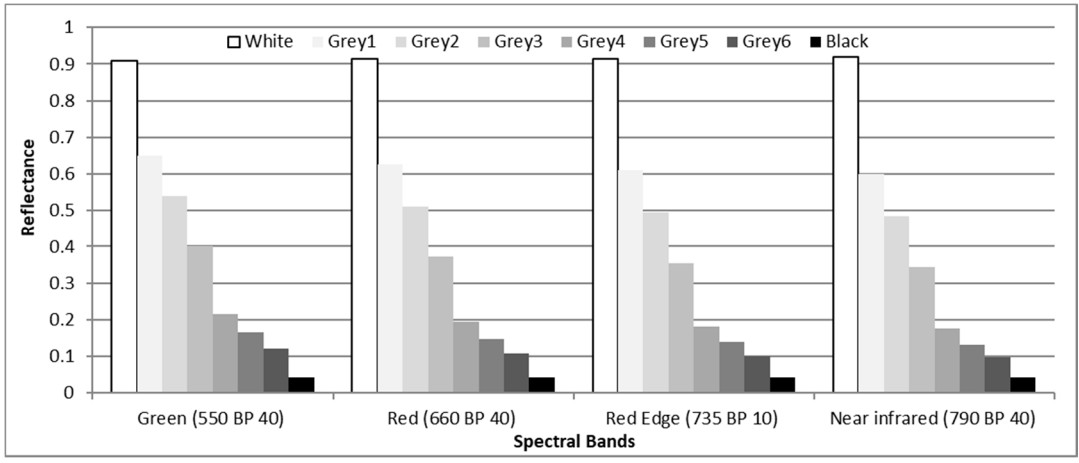

**Figure 3.** Spectral reflectance of the eight greyscale radiometric calibration panels measured in-situ with a field spectrometer and resampled to match the four Parrot Sequoia®multi-spectral bands.

### 2.4. Image Pre-Processing

Agisoft PhotoScan Pro (Agisoft LLC, St. Petersburg, Russia) was used to process the multi-spectral UAS data with the aid of an in-house Python script to optimise and automate the process. The key point and tie point limits were set as 40,000 and 10,000 respectively for photo alignment. The measured ground control points (GCPs) derived from the AeroPoints®were imported to geometrically calibrate and geo-reference the images. The geo-referencing root-mean-square error (RMSE) of the GCPs was 0.07 m. The overall projection error was 0.6 pixel based on the bundle-adjustment error assessment report. The projection errors of tie points were taken into account to eliminate poor-quality points, and a noise filter was also applied to assure the quality of the final products. A mild noise filter was applied during the point cloud densification process to keep as many details on the tree structure as possible. The densified photogrammetric point clouds were then classified to identify ground points before generating both a digital surface model (DSM) and digital terrain model (DTM). Before ortho-rectifying images, colour correction was enabled to calibrate the vignetting effect and reduce brightness variation caused by variations in photographic parameters (e.g. ISO value, shutter speed, etc.), which cannot be set to a constant value for the Parrot Sequoia camera. Orthomosaics were also produced with the colour balance option disabled to preserve the pixel values as close to the original images as possible [41,43] (Figure 4).

### 2.5. Producing Analysis Ready Data

In order to estimate the tree height, we derived a canopy height model (CHM) by subtracting the DTM from the DSM. For PPC, we used the brightness of red edge and NIR bands and a range of vegetation indices based on the at-surface reflectance imagery from the orthomosaic. The brightness is the direct output of image pre-processing, which was recorded in digital number (DN) and had a strong causality with at-surface reflectance. At-surface reflectance imagery was created using a simplified empirical correction based on the radiometric calibration panels, which was suggested by previous studies [7,41,42], with the aid of an in-house Python script. However, the initial corrected at-surface reflectance imagery appeared with some negative reflectance values due to shaded leaves and perhaps inaccurate reflectance estimations from the panel [41]. Therefore, we manually delineated the tree rows and calculated the zonal statistics within these regions to get the minimum value for each band of each mosaic. These negative values were applied for dark feature subtraction [35] to

offset the pixel values in the reflectance images to make sure that most of the at-surface reflectance values within the tree rows were positive and the spectral signature fitted the spectral characteristic of vegetation. Subsequently, the two at-surface reflectance orthomosaics were used to generate the following vegetation indices: (1) Normalised Difference Vegetation Index (NDVI), (2) Normalised Difference Red Edge Index (NDRE), (3) green NDVI (g-NDVI), (3) Red Edge Normalised Difference Vegetation Index (RENDVI), and (5) Green Chlorophyll Index (ClGE) (Table 2). These indices were selected because they are commonly used in precision agricultural applications and have proven useful for estimating the biomass and productivity with a specific degree of accuracy [12,18,36,44,45].

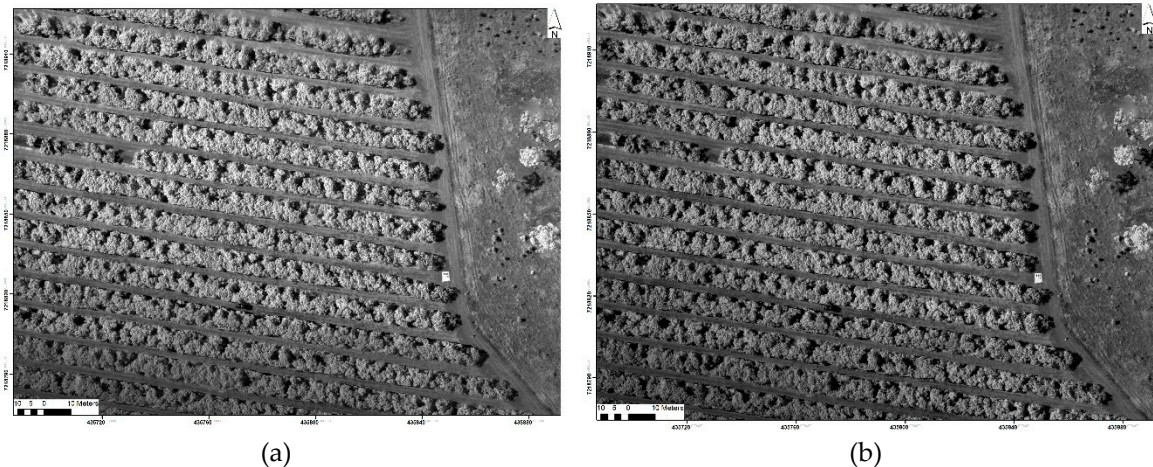

$$(a) \qquad\qquad (b)$$

**Figure 4.** The orthomosaics of the red edge band (**a**) without colour-balancing and (**b**) with colour-balancing. The south part of the mosaic in (**a**) is darker than the north part, while such brightness variation is less pronounced in (**b**). This phenomenon was caused by dynamic photographic parameters and only occurred in the green and red edge bands in this case.

**Table 2.** Formulae of vegetation indices which were used in this study.

| Vegetation Index | Formula |
| --- | --- |
| Normalised Difference Vegetation Index (NDVI) | $\frac{R_{NIR} - R_{Red}}{R_{NIR} + R_{Red}}$ |
| Green Normalised Difference Vegetation Index (gNDVI) | $\frac{R_{NIR} - R_{Green}}{R_{NIR} + R_{Green}}$ |
| Red Edge Normalised Difference Vegetation Index (RENDVI) | $\frac{R_{Red\ edge} - R_{Red}}{R_{Red\ edge} + R_{Red}}$ |
| Normalised Difference Red Edge Index (NDRE) | $\frac{R_{NIR} - R_{Red\ edge}}{R_{NIR} + R_{Red\ edge}}$ |
| Green Chlorophyll Index (ClGE) | $\frac{R_{NIR}}{R_{Green}} - 1$ |

$R_{band}$ represents the reflectance of specific bands.

## 2.6. Individual Tree Crown Delineation and Structural Attribute Estimation

To delineate individual trees, tree rows were first delineated with geographic object-based image analysis (GEOBIA) using the eCognition Developer software (Trimble, Munich, Germany). The initial segmentation was conducted using the multiresolution segmentation algorithm [46] based on the CHM, NDVI, NIR and red bands (Figure 5). The NDVI and NIR bands are commonly used to delineate vegetation [7,47,48]. The reason we also selected the red band as one of the inputs was that the ground material is mostly red dirt, creating contrast between trees and the ground. Initially, parts of the tree crowns were distinguished with segments that had CHM values greater than 3 m. Those segments were then grown progressively outwards until CHM $\leq$ 0.3 m and NDVI $\leq$ 0.1. At this stage, the delineated tree crown extent excluded significant within-crown gaps and left them as unclassified segments which were enclosed by tree crown segments. For gap fraction analysis purposes, such unclassified segments were included as part of the tree crown, if they were surrounded by already classified tree

crown objects. All the tree segments were eventually merged to create the extent shapefile of tree rows (Figure 5).

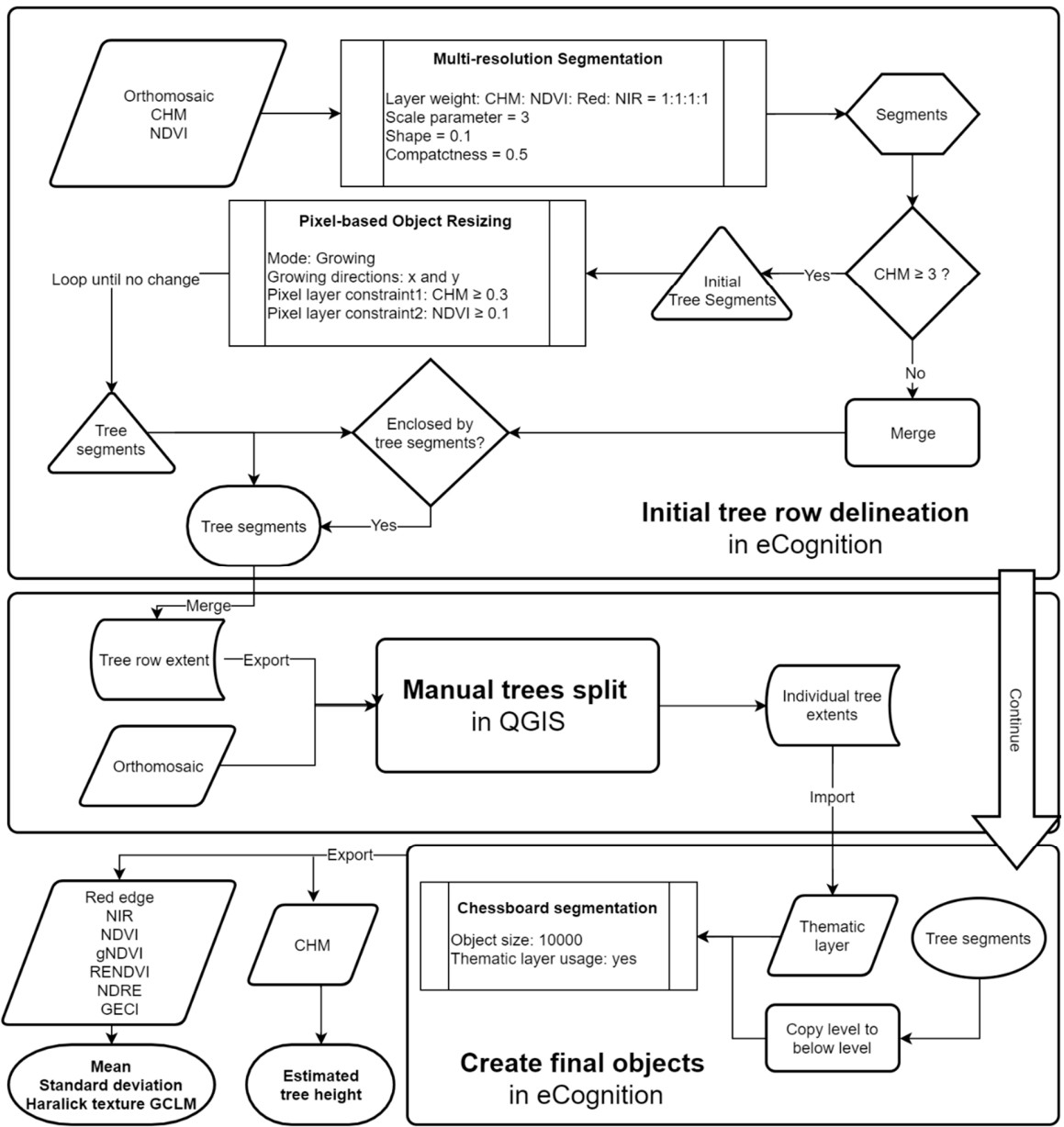

**Figure 5.** The workflow of individual tree delineation and attribute extraction.

Once the tree rows had been delineated, the next step was to identify individual tree crowns for the 45 selected trees, which had been measured in the field, as well as the 66 trees that had their condition ranked. Because the trees had been pruned mechanically and many of the tree crowns overlapped with the adjacent tree crowns, it was not possible to separate them automatically based on height and spectral information. Therefore, we visually delineated the 45 individual tree crowns based on visual interpretation of the orthomosaics and knowledge of tree location and planting distance. Some of the overlapping crown areas were split in half when the real extents were not visually distinguishable. Nevertheless, this method kept the majority of the extent of individual tree crowns, hence reducing bias in the comparison with field-derived PPC information. Although the proposed GEOBIA method could not further delineate individual neighbouring trees within tree rows, it reduced the potential error and processing time caused by manual delineation of tree row extent.

The semi-automatically delineated individual tree crown extents were used as the final objects to extract individual tree crown parameters. Tree height was estimated by the maximum CHM value within each object to correspond to field measurements. Crown width and length, which represented the horizontal diameter of the tree crown in the direction along and perpendicular to the hedgerows, respectively, was calculated using the width and length of the minimum rectangle that covered each crown extent (Figure 6a). Crown length for each tree was also manually measured from the orthomosaic for comparison with the calculated length using the GEOBIA method. The mean, standard deviation, and Haralick texture grey level co-occurrence matrix (GLCM) [7,46,49–52], including homogeneity, dissimilarity, contrast, and standard deviation, were extracted for the red edge brightness, NIR brightness, and five vegetation indices (Table 2) and exported for further analysis. These image-derived parameters were then compared to field measurements, and their coefficient of determination ($R^2$) of linear regressions and RMSE were calculated to assess their correlation and accuracy at the tree crown object level.

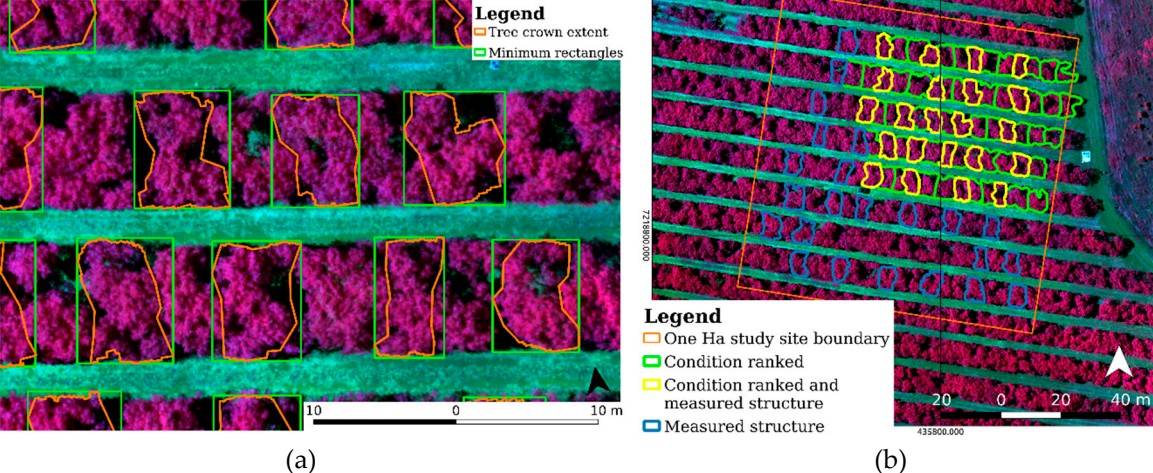

(a)          (b)

**Figure 6.** The delineated tree extent for the 45 selected trees that had in-situ structural measurements and the 66 trees that had on-ground condition evaluation. The rectangles which are highlighted in green in (**a**) represent the minimum rectangle to measure crown width and length, representing the horizontal distance in the direction along and perpendicular to the hedgerow, respectively. The yellow highlighted trees in (**b**) include the 22 trees for which tree crown were condition ranked and measured structure in field.

*2.7. Canopy Condition Ranking*

After finishing the accuracy assessment of the extracted structural attributes, only the variables such as the maximum CHM value, average pixel value and texture from spectral bands and vegetation indices, which were highly correlated with the field measurements, were selected for the condition ranking classification of the 66 field assessed trees (Figure 6b), using a parallel random forest classifier [53] with the aid of an in-house Python script. For the random forest classification, 80% of the 66 avocado trees were selected randomly to train the classifier with the aforementioned selected variables as training features and the condition ranking as training labels. Every time the classifier was trained, it randomly picked a number of training features that equalled the square root of total number of features to grow the tree. The accuracy of the trained classifier was estimated using out-of-bag error that was calculated by the bootstrap aggregating (bagging) predictor, which is the method for generating random subsets that have a uniform probability of each label and with replacement for prediction [53,54]. The classifier was trained repeatedly 300 times with different random subsets to allow nodes to grow until the out-of-bag error was stable [55]. The trained random forest classifier was then used to predict the condition ranking for all 66 trees based on the extracted variable sets. The prediction was compared with the in-field condition ranking to generate a confusion matrix

to assess the model accuracy. The relative importance of the features used as decision nodes was calculated by averaging the probabilistic predictions, which were estimated by the mean decrease impurity method, to decrease the variance as well as bias [56,57].

## 3. Results

### 3.1. Accuracy of CHM-Derived Tree Height

The maximum value of the UAS-derived CHM was extracted for the individual trees and compared against the field-based tree height measurements. The result shows that the CHM had a positive correlation with the field-derived tree height and produced a coefficient of determination ($R^2$) of 0.65 using a linear model (Figure 7). Of the 45 trees measured, the average field-derived tree height was 8.7 m (n = 45), while the average CHM-derived tree height was 8.4 m. The RMSE of the estimated tree height was 0.5 m. The average tree height was slightly underestimated using the CHM-derived method possibly due to the inaccuracies of 3D reconstruction on some branches.

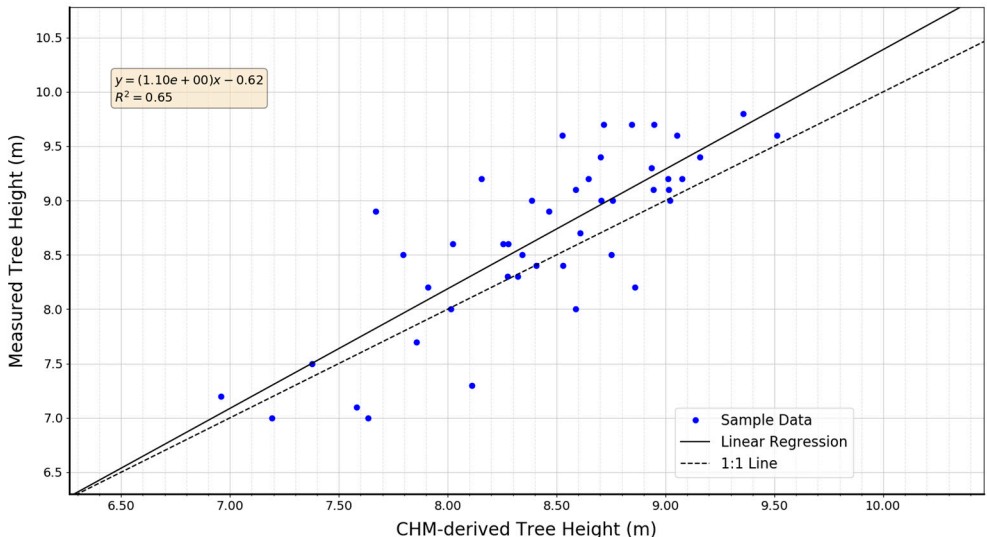

**Figure 7.** Scatter plot and the linear regression of tree height. Blue dots appearing above and below the 1:1 line represent a UAS-based under- and overestimation of the tree height, respectively.

### 3.2. Accuracy of Image-Derived Crown Width and Length

The correlation between the image-derived and field measured canopy width was low, because of the overlapping canopies and lack of height variation between them, preventing accurate delineation of the edges between neighbouring tree crowns from the CHM and spectral imagery. From Figure 8a, we can tell that there is no correlation between the measured canopy width and image-derived canopy width. Because of this limitation to estimate the canopy width accurately, it is not feasible to use the estimated canopy width as an indicator of tree condition.

The estimation of crown length using a linear model had a $R^2$ value of 0.88 compared to the manually measured length using the orthomosaic (Figures 6a and 8b). From Figure 8b, we can tell that there was an outlier in the scatter plot, which was caused by a branch of sparse leaves that failed reconstructed with SfM. Therefore, it was excluded from the statistical analysis. The average of the manually measured canopy length was 8.45 m (n = 45), while the average of the calculated canopy length based on the semi-automatic delineation results was 8.32 m. The calculated canopy length was underestimated for most of the trees with an RMSE at 0.2 m, because the leaves at the edge of canopy were usually sparse and hence not well-reconstructed with SfM. The tree canopies were pruned mechanically so that the edge of canopies in the direction along the hedgerow aligned very well. Hence, the canopy length is dictated by the pruning rather than the real condition of individual

tree crowns. While the pruning effects prevent measured length from being used as an indicator of tree condition, tree crown length and gap length between hedgerows can still provide useful information for future pruning operations and management purposes.

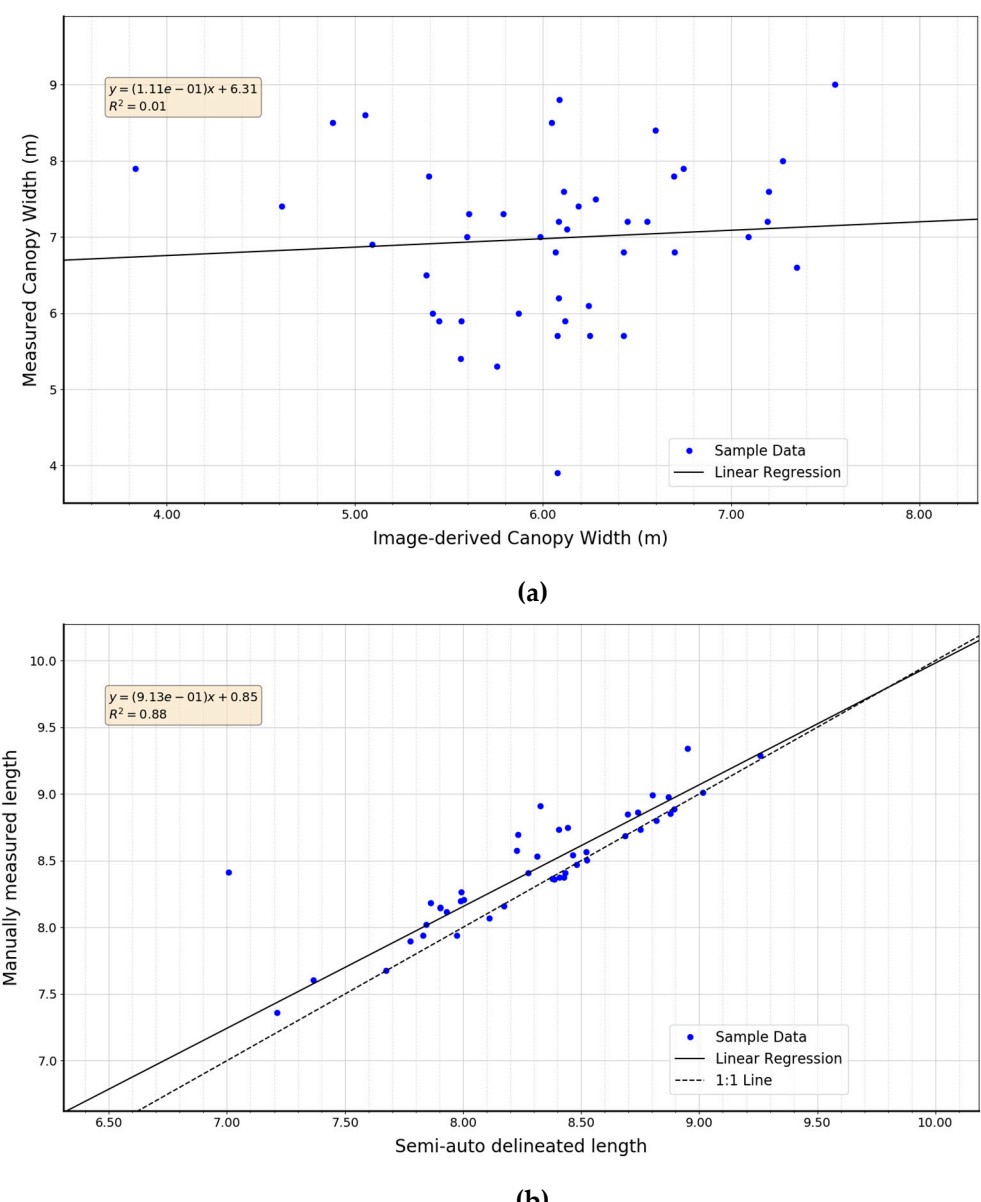

**Figure 8.** Scatter plot and linear regression of (**a**) canopy width in the direction along the hedgerow, and (**b**) canopy length in the direction perpendicular to the hedgerow. The outlier in (**b**) caused by a branch of sparse leaves at the lower part of a tree was not taken into account in the linear regression.

### *3.3. PPC Correlation*

Figure 9 shows the different $R^2$ values for the PPC regression between and field derived PPC values and different spectral indicators, including the brightness of the red edge and NIR bands and five vegetation indices and texture bands, with and without colour correction. Colour correction improved the observed relationship of image derived structure values to ground measurements. Comparing the results derived with and without colour correction, the $R^2$ value using the vegetation indices showed higher variation between the two radiometric corrections than the red edge and NIR band brightness. The PPC correlation among all the indicators was highest when integrating mean, standard deviation

(Stdev), and the four Haralick texture GLCMs for multi-variate linear regression. The input of Haralick texture GLCM improved the $R^2$ value more than the standard deviation, except for the use of the NIR brightness. The $R^2$ value using the mean of NIR brightness was 0.38 without colour correction but increased to 0.42 with colour correction. With the extra input of standard deviation and four Haralick texture GLCMs, the $R^2$ value increased to 0.59 and 0.61 for the NIR brightness using multi-variate linear regression without and with colour correction, respectively. When used to explain the variation in PPC, NDVI had the second highest $R^2$ value without colour correction and the highest $R^2$ of 0.62 with colour correction. The $R^2$ value using the mean NDVI value without colour correction was 0.31 and improved to 0.54 when both standard deviation and texture information were added. On the other hand, an $R^2$ of 0.56 was achieved using only the mean NDVI value with colour correction per tree crown, whereas this value was increased to 0.62 when including both standard deviation and texture information.

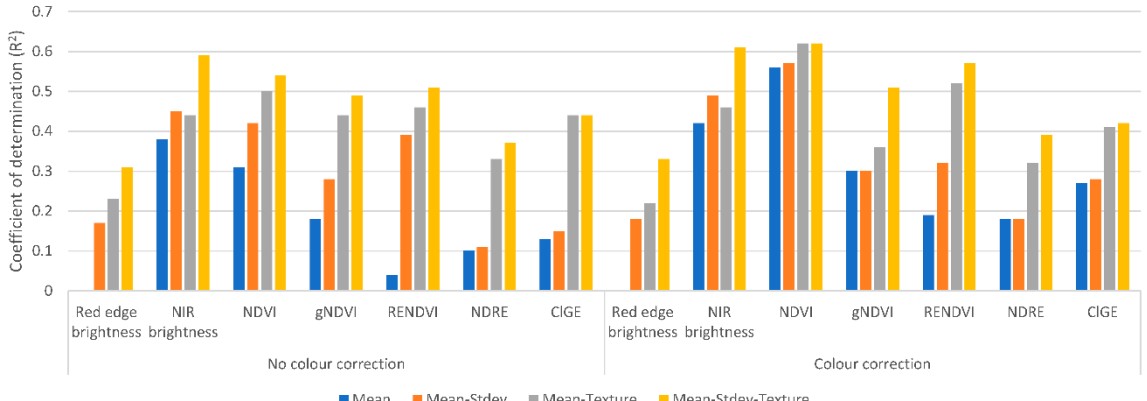

**Figure 9.** The $R^2$ value of the linear correlation between field measured PPC and different spectral images, including the brightness of the red edge and NIR bands and five vegetation indices. Each colour represents the different combination of input including mean, standard deviation (Stdev), and four texture GLCMs (Texture). The bars on the left and right sides of the graph were produced for the orthomosaics without and with colour correction, respectively.

### 3.4. Tree Condition Prediction

The mean, standard deviation, and four Haralick texture GLCMs of the NIR brightness and NDVI, derived from the colour-corrected orthomosaic, were extracted for the 66 trees. CHM-derived tree height was also selected as one of the features for predicting tree condition. The prediction results from both NIR and NDVI classifiers were identical (Figures 10 and 11). The seven-feature classifier had a higher tree condition prediction accuracy than the two-feature classifier for both the NIR brightness and NDVI. Both classifiers which used seven features had the accuracy of 98.48%, while the result based on two features produced an accuracy of 96.97%. In one and two cases for the seven and two-feature classifiers, respectively, the trees were mapped as moderate condition, while ranked in the field as fair condition.

The feature importance order was different between the NIR and NDVI classifiers (Tables 3 and 4). The variables enabling the greatest discrimination for both classifiers were the mean value, texture GLCM standard deviation, and the CHM-derived tree height, with their combination providing more than 50% probability of prediction. While the mean value was the top one feature for both the NIR and NDVI classifiers, the NDVI classifier relied on the mean value slightly more than the NIR classifier for predicting tree condition, which was 22.30% compared to 19.76%. The second and third most important features for the NIR classifier were the CHM-derived tree height (18.39%) and GLCM standard deviation (14.62%), respectively, while the order for the NDVI classifier was the opposite, with the GLCM standard deviation and CHM-derived tree height explaining 17.31% and 15.71% probability, respectively. In other words, the NIR classifier relied less on the spectral and

textural information compared to the NDVI classifier, which may be because the NIR brightness contains shadow-induced noise, which means that its derivatives may not properly reflect the actual canopy condition.

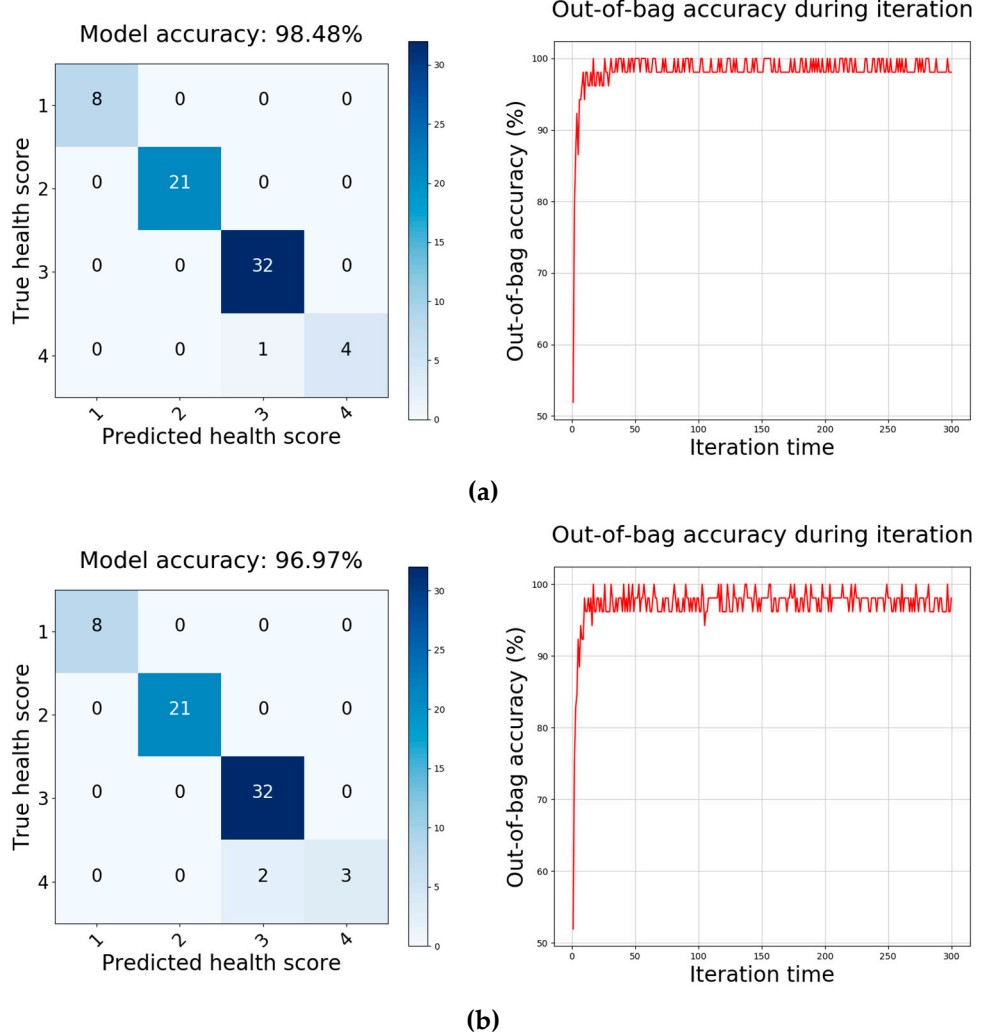

**Figure 10.** The confusion matrix and out-of-bag accuracy of the random forest classification for the tree condition prediction based on the CHM-derived tree height and (**a**) NIR-related features, including the mean, standard deviation, and four Haralick texture GLCMs; and (**b**) predicted PPC using all the six NIR-related features. The condition ranking from one to four represents condition from excellent to fair.

**Table 3.** The feature importance of each classifier using CHM-derived tree height and NIR-brightness-related features.

| Classifier | Feature | Feature Importance |
|:---:|:---:|:---:|
| | Mean NIR | 19.76% |
| | CHM-derived tree height | 18.39% |
| | GLCM Standard deviation | 14.62% |
| 7-features classifier (Figure 10a) | GLCM contrast | 12.77% |
| | GLCM dissimilarity | 12.06% |
| | Standard deviation | 11.59% |
| | GLCM homogeneity | 10.81% |
| 2-features classifier (Figure 10b) | CHM-derived tree height | 51.16% |
| | NIR-derived PPC | 48.84% |

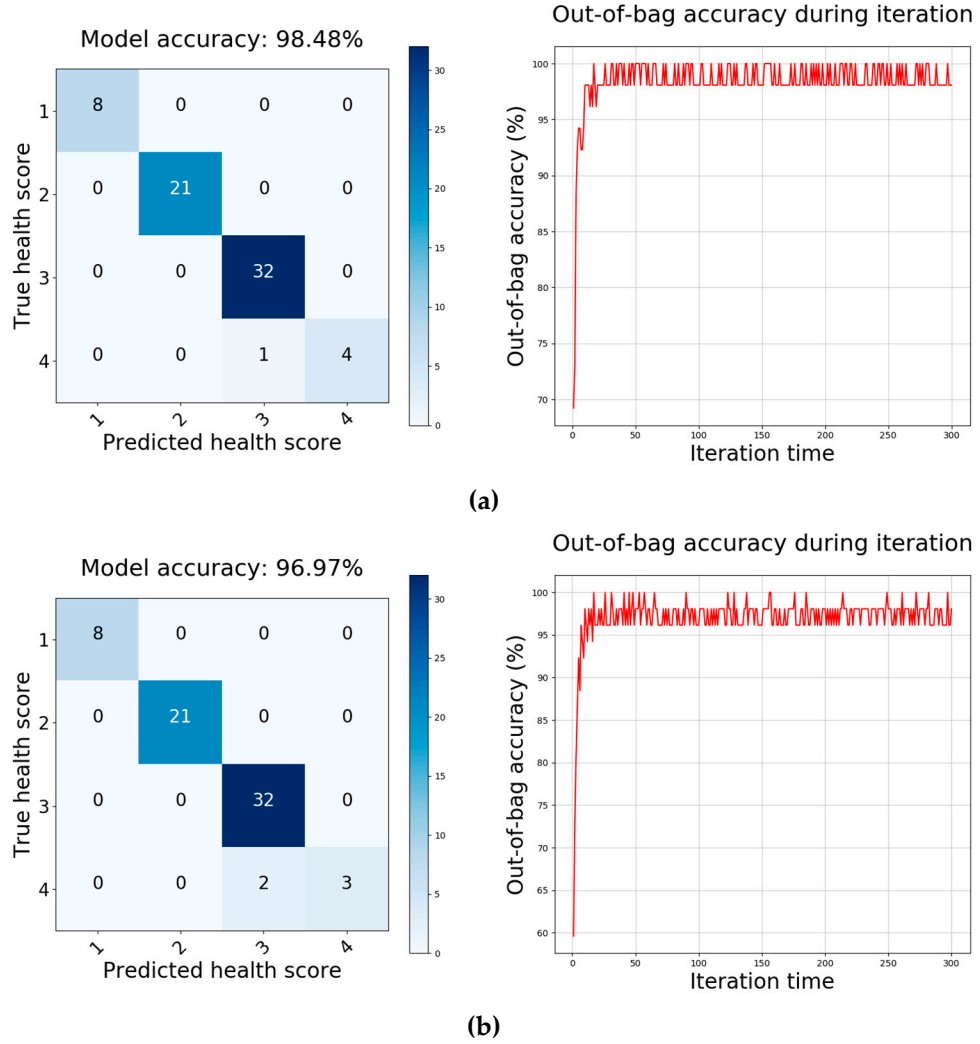

**Figure 11.** The confusion matrix and out-of-bag accuracy of the random forest classification for the tree condition prediction based on the CHM-derived tree height and (**a**) NDVI-related features, including the mean, standard deviation, and four Haralick texture GLCMs; and (**b**) predicted PPC using all the six NDVI-related features. The condition ranking from one to four represents condition from excellent to fair.

**Table 4.** The feature importance of each classifier using CHM-derived tree height and NDVI-related features.

| Classifier | Feature | Feature Importance |
|---|---|---|
| 7-features classifier (Figure 11a) | Mean NDVI | 22.30% |
| | GLCM standard deviation | 17.31% |
| | CHM-derived tree height | 15.71% |
| | Standard deviation | 12.96% |
| | GLCM homogeneity | 10.81% |
| | GLCM contrast | 10.75% |
| | GLCM dissimilarity | 10.16% |
| 2-features classifier (Figure 11b) | CHM-derived tree height | 51.44% |
| | NDVI-derived PPC | 48.56% |

## 4. Discussion

The average tree height was underestimated with the CHM-derived method. A similar phenomenon was also observed in previous studies of lychee and olive trees [7,8], and may be

attributed to inaccuracies in the 3D reconstruction of some branches that full of leaves in the SfM-generated point cloud [29]. The $R^2$ value was 0.65, which was very similar to previous studies conducted at similar flying altitude for lychee trees [7] and similar ground sample distance (GSD) on hedgerow olive trees [4]. Although the $R^2$ value showed a moderate correlation between the CHM-derived tree height and ground-measured tree height, it was significantly lower compared to the correlation between the estimated and measured tree height for olive trees in previous studies [6,8]. One reason could be differences in the configuration of UAS flight settings. In the study of Zarco-Tejada, Diaz-Varela, Angileri and Loudjani [6] on olive trees, the images were acquired using a grid flying pattern, which increased the parallax variation and the accuracy of the SfM 3D reconstruction. The second reason is the difference of canopy structure complexity. The canopy of a single avocado tree is usually formed by about 5 small branches that derivate from the main stem and alternatively become the major stem [10]. The more complex canopy structure of avocado trees compared to olive trees may have reduced the accuracy of the generated CHM. Nevertheless, the RMSE was 0.5 m and the relative RMSE in this study was around 6%, which was almost identical or even better than the relative error observed in other studies of olive trees. Previous studies suggested that higher image spatial resolution could lead to higher $R^2$ of tree height estimation [6–8]. Besides, studies also argued that lower flight altitude may improve the $R^2$ even if the spatial resolutions were almost identical [4,8]. This phenomenon may because higher parallax variation increased the visibility of tie points when acquiring images at a lower altitude [7,34]. However, other studies have also suggested to acquire UAS data at approximate 80 m AGL to prevent serious bidirectional reflectance distribution function (BRDF) effect and achieve a balance between data acquisition efficiency and quality for measuring trees [40,41].

The highest $R^2$ value of the PPC regression occurred when combining the mean, standard deviation and Haralick texture GLCMs, including homogeneity, standard deviation, dissimilarity, and contrast. It is known that the reflectance of visible light is influenced significantly by the concentration of photosynthetic pigments, while the reflectance of NIR is sensitive to the leaf and canopy structure of vegetation [34–36]. Unlike a previous study of lychee trees, where the use of mean NIR brightness explained 80% of the variability of PPC [7], the achieved $R^2$ based on mean NIR brightness was only 0.38 and 0.42 without and with colour correction, respectively. We suspected that the more complex canopy structure of avocado tree crops contributed to the lower $R^2$ value. Based on the 'rhythmic' canopy expansion pattern of lychee trees compared to the higher trunk-branch differentiation of avocado trees [31,32], it is likely that the different types of canopy geometric characteristic had different influences on the spectral reflectance that observed by the sensor. Studies on the radiometric correction for multi-spectral UAS imagery show that complex canopy structure produce more diffuse radiant flux within the canopy [41,58]. Therefore, we suspect that the observed reflectance of avocado trees was less representative of the actual reflected radiant energy due to the aforementioned reason, causing the lower correlation between the observed reflectance and PPC. Figure 9 shows that a smaller improvement in correlation was achieved for NIR brightness than that of NDVI by just adding texture GLCMs. The shadow-induced noise in NIR brightness may be due to less accurate PPC prediction using only NIR mean and texture information (Figure 9). Comparing NIR and NDVI to the other indicators, the remaining indicators had fewer issues with saturation due to high canopy density and were more sensitive to pigment concentration variation [36,44,45,59]. Surprisingly, the saturation drawback of NIR and NDVI turned out to be the advantage for better PPC estimation for avocado trees.

The radiometric correction of the multi-spectral UAS imagery played an important role in the PPC estimation. As mentioned in Figure 4, brightness variation in the green and red edge bands was observed because of the dynamic photographic parameter settings of the camera. Therefore, the estimated reflectance would have been less accurate if such variation had not been addressed, which would have affected the calculated vegetation indices. The $R^2$ values of the PPC regression with vegetation indices were all higher after reducing the brightness variation with colour balancing

(Figure 9). However, to radiometrically correct the image brightness to accurate reflectance, previous result showed that effective BRDF correction, accurate sensor-information-based calibration, and simultaneous irradiance measurement are all potential procedures that can be considered [41,58]. Based on our study, we suggest that if the image is radiometrically corrected, the multi-variate linear regression of mean, standard deviation, and four Haralick texture GLCMs of NDVI could be a strong indicator of PPC over NIR brightness, as it provided the highest correlation with field-derived PPC. Nevertheless, radiometric correction of UAS imagery has proven to be a sophisticated process and sometimes difficult for delivering accurate results [41,58,60].

The proposed GEOBIA workflow showed its limitation to accurately delineate individual tree crown extents for avocado hedgerows. Although a previous study suggested that there was a strong correlation between measured and image-derived canopy diameter for hedgerow olive trees [4] and mango trees [13], the semi-automatically delineated canopy width in the direction along the hedgerow did not match the ground measurement in our study area, because of adjacent overlapping tree crowns with limited height variation between the tree apex and crown perimeter caused by mechanical pruning. Such delineation limitation occurs very often when applying remote sensing data in a hedgerow avocado orchard [61]. Therefore, it is necessary to make some assumptions such as a fixed distance between two adjacent trees to delineate individual tree crowns from a UAS orthomosaic. The semi-automatically delineated canopy length in the direction perpendicular to the hedgerow proved to be accurate. However, the canopy length is mainly controlled by pruning practices, and therefore, not suitable for tree condition assessment. Nevertheless, our study also showed that as long as the delineated tree crown extent contains the majority of the realistic tree crown extent, tree height can still be predicted and attributes, which include mean value, standard deviation, and texture GLCMs that are highly correlated to PPC, can be extracted for predicting tree condition. Alternatively, having an accurate GPS layer of individual tree centroids can assist with the delineation of individual trees. Most tree planting undertaken in recent times occur with GPS guidance and as such, these layers are readily available [12,13].

Our results showed that the estimated tree height and canopy spectral and textural attributes could be used as explanatory variables to predict the condition of avocado trees. Both the NIR and NDVI classifiers with seven feature inputs yielded accurate results for predicting avocado tree condition. The accuracy of both the NIR and NDVI classifiers were identical, though the NIR classifier had less variability using spectral and texture information for predicting tree condition. Considered the difficulty of applying accurate radiometric correction on multi-spectral UAS imagery, this characteristic may turn out to become a benefit as the variation of correlation between the NIR brightness derivatives and PPC is less influenced by the radiometric correction (see Figure 9). The seven-feature NIR classifier is recommended at this stage. It is noted that our data was acquired in the pre-pruning period, when tree structure was less influenced by previous limb removal. The tree height is usually controlled for management purposes. Therefore, the importance of tree height for condition ranking in different growing stages may be different. The ranking result in fair-conditioned trees had two trees ranked as fair, where were incorrectly classified as moderate. In the study of Salgadoe, Robson, Lamb, Dann and Searle [33] on root rot disease severity ranking for avocado trees, the results showed that the lower ranked trees had less differences in PPC and VI values. Hence, smaller PPC and spectral variation between different condition categories may cause mis-classification for lower ranked trees. Such mis-classification may cause farmers to miss the opportunity to apply appropriate treatments on a tree with fair condition, which might result in further tree condition degradation. It is therefore imperative that farmers are well informed about how UAS-based tree condition mapping approaches work and what their limitations might be to ensure appropriate interpretation and evaluation of results.

## 5. Conclusions

The study identified the accuracy of UAS-based mapping of avocado tree height and GEOBIA-derived canopy dimensions, as well as the correlation between PPC and both spectral

and textural attributes. The CHM-derived tree height and the spectral and textural attributes were used to rank the tree condition using a seven-feature NIR random forest classifier and achieved a 98.48% accuracy. Such techniques could potentially provide essential information for the horticulture industry to achieve plant-scale precision agricultural orchard management. Radiometric correction of UAS imagery is essential as a prerequisite for accurate PPC estimation using NDVI or any other spectral information. Future research should focus on testing condition ranking methods over larger areas to scale up to the orchard level. It would also be important to include more sampled trees with all four condition categories to train the classifier to more accurately meet the avocado industry's tree condition ranking standards in Australia. Furthermore, such condition ranking technique needs to be tested in different phenological stages to assess the robust features for condition ranking and its all-season performance. Eventually, the condition ranking method might be tested and adapted to different horticultural tree crops. By understanding the plant-scale tree condition, the farmers could potentially estimate the general tree crop productivity to develop better management strategies by applying appropriate agricultural treatments accordingly. Such condition ranking strategy could play a role to bridge the gap between the remote sensing expertise and farmers' knowledge of their tree crops.

**Author Contributions:** Conceptualization, all the authors; Methodology, Y.-H.T., S.P., and K.J.; Software, Y.-H.T.; Validation, Y.-H.T.; Formal Analysis, Y.-H.T.; Investigation, Y.-H.T. and K.J.; Resources, all the authors; Data Curation, Y.-H.T.; Writing-Original Draft Preparation, Y.-H.T.; Writing-Review & Editing, all the authors; Visualization, Y.-H.T.; Supervision, S.P., K.J., and A.R.; Project Administration, S.P.; Funding Acquisition, S.P. and A.R.

**Funding:** This research was funded by Department of Agriculture and Water Resources, Australian Government as part of its Rural R&D for Profit Program's subproject "*Multi-Scale Monitoring Tools for Managing Australia Tree Crops - Industry Meets Innovation*".

**Acknowledgments:** The authors would like to acknowledge the support from local farmers, especially Chad Simpson; fieldwork assistance from Dan Wu; and technical supports from online forums.

**Conflicts of Interest:** The authors declare no conflict of interest.

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
