# Peer review of "Measuring Canopy Structure and Condition Using Multi-Spectral UAS Imagery in a Horticultural Environment"

_remotesensing, doi:10.3390/rs11030269_

Round 1

Reviewer 1 Report

 Good study and paper on using UAV based multispectral imagery to estimate tree canopy cover and health condition. A few questions and improvements before publication:

·         Was there a reason to use “NIR brightness” rather than “NIR reflectance”?

·         The samples sizes were confusing. Suggest a figure/map better explain the three different sample sizes mentioned in the manuscript: 66, 45, 22. And since you used the tree structure (sample size 45) to evaluate the tree condition (sample size 66), why did you only have 22 overlapped samples in your study rather than all of the 45 trees? And was your estimate on the tree condition using tree structure only based on the 22 trees?

·         Line 210-212: “The unclassified segments which were enclosed by tree crown segments were included as tree crown for gap fraction analysis.”, what does it mean?

·         Line 202-224: not quite clear about what was done by the automatic segmentation by eCognition and what was done by the manual segmentation. If the individual trees were manually segmented, why to use eCognition to first to identify tree rows? It is fine for you to conclude that the manual segmentation works better because it is challenging to do it when the canopy is close. Just state it clearly.

·         Figure 6: cannot read the legend text and scale bar.

·         Define on how the tree crown width and length were measured, such as the width and length of each rectangular box.

·         Figure 10: define the “health score” in the figure caption. Is “1” the best health condition or the worst? Also, please increase the front size for all sub-figures.

·         Table 3 and 4: was there a reason why red edge band was not included in the models but only the NIR band? And are the “NDVI-derived PPC” and “Mean NDVI” highly correlated?

·         Line 389-395: did you investigate the red edge band which supposed to have better performance than NDVI after canopy closure?

Author Response

Point 1: Was there a reason to use “NIR brightness” rather than “NIR reflectance”?

Response 1: The ‘brightness’ represents the digital number of images after pre-processing and has a strong causality with ‘reflectance’. NIR brightness has proved to be correlated with the tree crown condition in a previous study, and it can save the trouble of radiometric correction. This point was addressed in the introduction and discussion section. We added the definition of ‘brightness’ in L 183-186.

Point 2: The samples sizes were confusing. Suggest a figure/map better explain the three different sample sizes mentioned in the manuscript: 66, 45, 22. And since you used the tree structure (sample size 45) to evaluate the tree condition (sample size 66), why did you only have 22 overlapped samples in your study rather than all of the 45 trees? And was your estimate on the tree condition using tree structure only based on the 22 trees?

Response 2: The 45 trees were evenly distributed in our study site for structure validation, whereas 66 trees were included within the same area for the condition assessment because of higher condition variability. We used the 45 samples to only confirm that there are correlations between the image-derived structures and on-ground structure measurement. For the 66 trees, we validated the mapping results against field-assessed condition. The 22 overlapping samples, i.e. trees that had both their structure and condition measured in the field, did not serve any purpose for the validation. To clarify this, we have added more description in L 107 and 126.

Point 3: Line 210-212: “The unclassified segments which were enclosed by tree crown segments were included as tree crown for gap fraction analysis.”, what does it mean?

Response 3: Some significant within-crown gaps were initially not classified as tree crown so to avoid that they incorrectly remained ‘unclassified’, an additional step in the rule set ensured that they were included if completely surrounded by objects classified as tree crown. This ensured that they could be used for the gap fraction analysis. Added more description in L 212-216.

Point 4: Line 202-224: not quite clear about what was done by the automatic segmentation by eCognition and what was done by the manual segmentation. If the individual trees were manually segmented, why to use eCognition to first to identify tree rows? It is fine for you to conclude that the manual segmentation works better because it is challenging to do it when the canopy is close. Just state it clearly.

Response 4: Although the proposed GEOBIA method could not further delineate individual neighbouring trees within tree rows, it reduced the potential error and processing time caused by manual delineation of tree row extent. Therefore, we chose to conduct the delineation in a semi-automatic way instead of fully-manual delineation. Added more description in L 228-231.

Point 5: Figure 6: cannot read the legend text and scale bar.

Response 5: Enlarged the map elements in the figures.

Point 6: Define on how the tree crown width and length were measured, such as the width and length of each rectangular box.

Response 6: It is described in L 234-237.

Point 7: Figure 10: define the “health score” in the figure caption. Is “1” the best health condition or the worst? Also, please increase the front size for all sub-figures.

Response 7: Added the descriptions and altered Figures 10-11.

Point 8: Table 3 and 4: was there a reason why red edge band was not included in the models but only the NIR band? And are the “NDVI-derived PPC” and “Mean NDVI” highly correlated?

Response 8: The reason we used only NIR brightness and NDVI is because they had the highest correlation with ground-measured PPC, which is described in L 248-251. In the introduction, we stated that Salgadoe, et al. [33] discovered that PPC is highly correlated with tree condition rankings. Therefore, it was assumed that the features which were highly correlated to PPC should be able to predict condition. Added more description in L 252. The NIR and NDVI-derived PPC were predicted using all the six features including mean, standard deviation, and four GLCMs, which were highly correlated with ground-measured PPC. Added some description in the caption of Figures 10 and 11.

Point 9: Line 389-395: did you investigate the red edge band which supposed to have better performance than NDVI after canopy closure?

Response 9: Yes, we did. The result, however, was not as good as those using NIR and NDVI, which is shown in Figure 9.

Reviewer 2 Report

The authors present a straight forward approach identified the accuracy of UAS-based mapping of avocado tree height and GEOBIA derived canopy dimensions, as well as the correlation between PPC and both spectral and textural attributes. The manuscript is coherent and intelligible. Unfortunately, I do not see much novelty in the presented work. To my opinion, the presented research doesn’t add much to existing knowledge. It is a good applied work.

The novelty may appear, for example, if the new/improved method for UAV data processing is used or/and exisitng methods are applied to another kind of RS data (airborne, high resolution satellite) and compared with UAV data.

Minor comments:

30: please, do not repeat title words in the keywords.

95: more suitable to Methods.

97: more suitable to Discussion.

130: brief description of condition categories is needed.

199: more suitable to Intro

To train the classifier more sampled trees with all four condition categories are recommended.

Author Response

Point 1: The manuscript is coherent and intelligible. Unfortunately, I do not see much novelty in the presented work. To my opinion, the presented research doesn’t add much to existing knowledge. It is a good applied work. The novelty may appear, for example, if the new/improved method for UAV data processing is used or/and existing methods are applied to another kind of RS data (airborne, high resolution satellite) and compared with UAV data.

Response 1: Indeed, most of the methods for structural measurements are mature. The focus and contribution of this study is to apply the UAV image-derived structural measurements to estimate the tree condition to match the industry’s need, which has not been previously done. Hence, the application of UAV imagery for mapping the condition of individual avocado tree crowns is novel. Therefore, we do not agree with this comment made by Reviewer 2. We consider this bridging knowledge to be important to fill the gap of knowledge by integrating the expertise between remote sensing and horticultural tree condition assessment.

Point 2: Line 30: please, do not repeat title words in the keywords.

Response 2: We removed the ‘multi-spectral’ keyword.

Point 3: Line 95: more suitable to Methods.

Response 3: We decided to remove this sentence for better cohesion.

Point 4: Line 97: more suitable to Discussion.

Response 4: We decided to alter some of the words and keep it there for better cohesion and to emphasise this concept as this is this paper’s focus and contribution.

Point 5: Line 130: brief description of condition categories is needed.

Response 5: Added the description in the caption of Figure 2.

Point 6: Line 199: more suitable to Intro

Response 6: This sentence was used to explain the reason why we selected these indices and would be more appropriate to put in this section. We altered some words to make it more suitable for the Methods section.

Point 7: To train the classifier more sampled trees with all four condition categories are recommended.

Response 7: This is indeed important future work and we also address this issue in L 479.

Round 2

Reviewer 2 Report

Dear Authors,

thank you for your answers.